# Apolipoprotein E4 Is Associated with Right Ventricular Dysfunction in Dilated Cardiomyopathy—An Animal and In-Human Comparative Study

**DOI:** 10.3390/ijms22189688

**Published:** 2021-09-07

**Authors:** Rodica Diaconu, Nicole Schaaps, Mamdouh Afify, Peter Boor, Anne Cornelissen, Roberta A. Florescu, Sakine Simsekyilmaz, Teddy El-Khoury, David Schumacher, Mihai Ioana, Ioana Streata, Constantin Militaru, Ionut Donoiu, Felix Vogt, Elisa A. Liehn

**Affiliations:** 1Human Genetic Laboratory, Faculty of Medicine, University of Medicine and Pharmacy, 200349 Craiova, Romania; rodica.diaconu91@gmail.com (R.D.); mihai.ioana@umfcv.ro (M.I.); ioana.streata@yahoo.com (I.S.); 2Department of Cardiology, University of Medicine and Pharmacy, 200349 Craiova, Romania; cccmilitaru@yahoo.com (C.M.); i.donoiu@gmail.com (I.D.); 3Department of Cardiology, Angiology and Intensive Care, Medical Faculty, RWTH Aachen University, 52074 Aachen, Germany; nschaaps@ukaachen.de (N.S.); mafify@ukaachen.de (M.A.); acornelissen@ukachen.de (A.C.); rflorescu@ukaachen.de (R.A.F.); telkhoury@ukaachen.de (T.E.-K.); fvogt@ukaachen.de (F.V.); 4Department of Pathology, Faculty of Veterinary Medicine, Cairo University, Giza 12211, Egypt; 5Medical Faculty, Institute of Pathology, RWTH Aachen University, 52074 Aachen, Germany; pboor@ukaachen.de; 6Department for Pharmacology and Clinical Pharmacology, Medical Faculty, University Hospital Düsseldorf, 40225 Düsseldorf, Germany; sakine@gmx.de; 7Department of Anesthesiology, Medical Faculty, RWTH Aachen University, 52074 Aachen, Germany; dschumacher@ukaachen.de; 8Medical Faculty, Institute of Experimental Medicine and Systems Biology, RWTH Aachen University, 52074 Aachen, Germany; 9Medical Faculty, Institute for Molecular Cardiovascular Research (IMCAR), RWTH Aachen University, 52074 Aachen, Germany; 10Institute for Pathology “Victor Babes”, 050096 Bucharest, Romania

**Keywords:** apoE, cardiovascular models, dilated cardiomyopathy, right ventricular systolic dysfunction, ε4 allele

## Abstract

ApoE abnormality represents a well-known risk factor for cardiovascular diseases. Beyond its role in lipid metabolism, novel studies demonstrate a complex involvement of apoE in membrane homeostasis and signaling as well as in nuclear transcription. Due to the large spread of apoE isoforms in the human population, there is a need to understand the apoE’s role in pathological processes. Our study aims to dissect the involvement of apoE in heart failure. We showed that apoE-deficient rats present multiple organ damages (kidney, liver, lung and spleen) besides the known predisposition for obesity and affected lipid metabolism (two-fold increase in tissular damages in liver and one-fold increase in kidney, lung and spleen). Heart tissue also showed significant morphological changes in apoE^−/−^ rats, mostly after a high-fat diet. Interestingly, the right ventricle of apoE^−/−^ rats fed a high-fat diet showed more damage and affected collagen content (~60% less total collagen content and double increase in collagen1/collagen3 ratio) compared with the left ventricle (no significant differences in total collagen content or collagen1/collagen3 ratio). In patients, we were able to find a correlation between the presence of ε4 allele and cardiomyopathy (χ^2^ = 10.244; *p* = 0.001), but also with right ventricle dysfunction with decreased TAPSE (15.3 ± 2.63 mm in ε4-allele-presenting patients vs. 19.8 ± 3.58 mm if the ε4 allele is absent, *p* < 0.0001*) and increased in systolic pulmonary artery pressure (50.44 ± 16.47 mmHg in ε4-allele-presenting patients vs. 40.68 ± 15.94 mmHg if the ε4 allele is absent, *p* = 0.0019). Our results confirm that the presence of the ε4 allele is a lipid-metabolism-independent risk factor for heart failure. Moreover, we show for the first time that the presence of the ε4 allele is associated with right ventricle dysfunction, implying different regulatory mechanisms of fibroblasts and the extracellular matrix in both ventricles. This is essential to be considered and thoroughly investigated before the design of therapeutical strategies for patients with heart failure.

## 1. Introduction

Apolipoprotein (apo) E abnormality and disturbed lipid transportation represent well-known risk factors for cardiovascular diseases [1,2,3]. However, new studies demonstrate a complex involvement of apoE in membrane homeostasis and signaling as well as in nuclear transcription [4]. Due to the large spread of apoE isoforms in the human population, there is a need to understand apoE’s role in pathological processes to explain the current therapeutic failure in some patients and to develop proper personalized therapeutic strategies.

There are six different *apoE* genotypes: three homozygotes (ε2ε2, ε3ε3, ε4ε4) and three heterozygotes (ε2ε3, ε3ε4, ε2ε4). The ε3ε3 genotype is considered wild-type [5], as it is the most common with a prevalence of 62% in the population [4]. ε4 allele carriers seem to have the highest risk for coronary heart disease and stroke, being equivalent to a dysfunctional *apoE* gene and imitating the genetical deletion in animal models. Besides the risk for developing atherosclerosis, ε4 allele carriers also seem to develop dilated cardiomyopathy [6]. Dilated cardiomyopathy is a progressive disease consisting of heart wall thinning and a gradual increase in left ventricular end-diastolic and end-systolic volumes, evolving towards heart failure with reduced ejection fraction and representing the third cause of heart failure in regard to frequency and the most common disease for which cardiac transplantation is required [7].

Since the involvement of apoE in the pathology of dilated cardiomyopathy is still not known, the aim of the study is to investigate the lipid-independent effects of apoE on heart tissue and the possible correlation with non-ischemic heart pathology in animals and humans.

## 2. Results

Time- and diet-dependent changes in blood composition in apoE knockout (apoE^−/−^) rats were analyzed.

Wild-type and apoE^−/−^ rats were fed a normal or high-fat diet over 12 weeks. Blood analyses were performed at the indicated time points. As expected, a high-fat diet induced a significant increase in the weight of the rats; however, this was seen in both wild-type and apoE^−/−^ rats (Figure 1A). Further, the high-fat diet induced a significant increase in blood lipids, such as cholesterin (Figure 1B), low-density lipoprotein (LDL, Figure 1D) and high-density lipoprotein (HDL, Figure 1E) only in apoE^−/−^ rats, while in apoE^−/−^ fed a normal diet and wild-type rats, these lipids remained unchanged. Another pattern was seen in triglyceride concentrations (Figure 1C). In wild-type rats, the high-fat diet induced a significant triglyceride concentration increase over time, whereas in apoE^−/−^ rats, the triglyceride concentration was increased at a constant level. Further, the high-fat diet induced increase in acid uric (Figure 1H) in both groups over time; however, only apoE^−/−^ rats showed a deterioration of renal function, as determined by increased creatinine concentration (Figure 1F). Albumin (Figure 1G), glucose (Figure 1I) and blood cells, such as total leukocytes (Figure 1J), lymphocytes (Figure 1K), monocytes (Figure 1L), granulocytes (Figure 1M), erythrocytes (Figure 1N) and platelets (Figure 1P), were not affected by genotype or the high-fat diet.

### 2.1. Morphological Changes Induced by High-Fat Diet in Different Organs

Stained sections from different organs were analyzed regarding the cellular and morphological changes induced by the high-fat diet. Almost all organs presented pathological changes in old apoE^−/−^ rats, independently of diet.

The liver was most susceptible and presented signs of hepatocellular vacuolization, fat deposition and focal necrosis after the high-fat diet in both wild-type and apoE^−/−^ rats (Figure 2A) after feeding of the high-fat diet. However, the pathological changes were significantly increased in apoE^−/−^ rats compared with the wild-type rats.

Independently of diet type, the kidneys of apoE^−/−^ rats presented increased signs of vacuolization, congestion and glomerular membrane thickening compared with wild-type rats (Figure 2B). Similarly, we found more edema, congestion and thickness of the alveolar wall in the lung (Figure 2C) of apoE^−/−^ rats, while the spleen (Figure 2D) showed more lymphoid hyperplasia and congestion in apoE^−/−^ rats compared with wild-type rates.

No changes were observed in intestine morphology (Figure 2E) in wild-type and apoE^−/−^ rats fed or not fed a high-fat diet.

### 2.2. Heart Morphological Changes in ApoE^−/−^ Fed or Not High-Fat Diet

Heart tissue was analyzed separately in the left (Figure 3A) and right ventricle (Figure 3B). Both ventricles presented loose interstitial tissue (Figure 3C) in apoE^−/−^ rats fed the high-fat diet, suggesting defects in the extracellular matrix in preserving the structural integrity. Similarly, we detected increased areas of necrosis (Figure 3D) in both ventricles in apoE^−/−^ rats after feeding the high-fat diet, suggesting more susceptible myocardium in this group. No differences in the vacuolization of the cells were observed (Figure 3E). Further, the right myocardium of apoE^−/−^ rats fed the high-fat diet showed increased bundles of thin tissue (Figure 3F) and mononuclear cell infiltration (Figure 3G), while the left myocardium showed more prominent edema (Figure 3H).

### 2.3. Histomorphometrical Characterization of Hearts in ApoE^−/−^ Rats Fed with Normal or High-Fat Diet

Since many studies point out the role of apoE^−/−^ in the development of dilatative cardiomyopathy, we performed histological measurements of heart cavities and walls. While wall thickness presented a slight tendency to decrease (Figure 4A) and ventricular cavity to increase (Figure 4B) in apoE^−/−^ rats fed the high-fat diet, the measurements were not significant either in the left or right ventricle.

Collagen content was only significantly decreased in the right ventricle of rats fed the high-fat diet, and with an apoE^−/−^ background (Figure 4C,D), while no significant changes were observed in the left ventricle. Similarly, the collagen1/collagen3 ratio was only increased in the right ventricle of apoE^−/−^ rats, while in the left ventricle it remained unchanged (Figure 4E).

### 2.4. Study Group Population and Characteristics

For the validation of the results from the animal study, we further investigated the correlation between the human *apoE* allele with dimensional and functional heart parameters.

For this purpose, a total of 111 consecutive Caucasian patients (85 males) with advanced heart failure (mean age 66 ± 8.68 years) were included: 50 with ischemic cardiomyopathy (ICM) and 61 with dilated cardiomyopathy (DCM) (Figure 5A). The diagnosis was based on echocardiography and coronary angiogram. The mean left ventricular ejection fraction was 28.3 ± 6%, and the mean NYHA class was 2.93 ± 1.2. A total of 187 (131 males) controls (mean age 57 ± 10.3 years) were selected from healthy volunteers (Figure 5B). The main characteristics of the cohorts are presented in Table 1.

### 2.5. Comparison of Genotype Frequency and Allele Frequency between Groups

There was a significant difference in *apoE* genotype between CM patients and the control groups (Table 2 and Figure 6A). We found the ε4 allele frequency to be significantly higher in the CM patients group compared to the control (χ^2^ = 10.244; *p* = 0.001). CM patients presented a significant increase in the ε3/ε4 genotype frequency compared to the controls (χ^2^ = 11.012; OR = 2.31, 95% Cl = 1.404-3.831; *p* < 0.001). Further, the frequency of allele ε2 was higher in the CM patients (χ^2^ = 4.349; *p* = 0.03), whereas as expected, the frequency of the ε3/ε3 genotype was found to be significantly higher in the controls compared to patients (χ^2^ = 16.069; *p* = 0.0001).

Comparing the frequency of different genotypes inside the CM patients group, there were no significant differences between the patients with ICM or DCM (Table 3). The presence of the ε2/ε3 genotype was more frequent in the group of patients with ICM (*p* = 0.019) (Table 3, Figure 6B). However, due to the small number of patients presented these alleles (eight patients for ICM and two patients for DCM) this result would need confirmation with higher patient numbers.

### 2.6. The Main Characteristics of the Patient According to ε4 Allele Presence

Since the ε4 allele seems to correlate with increased cardiovascular events [4], we analyzed the influence of the ε4 allele on heart function. LVEF was significantly reduced in the presence of the ε4 allele compared to the patients without the ε4 allele. Moreover, the presence of the ε4 allele was associated with significantly increased pulmonary hypertension and dysfunctional right ventricle compared to the patients without the ε4 allele (Table 4).

There were no significant differences between the functional parameters between different genotypes; however, the ε2/ε4 genotype showed a non-significant trend towards reduced LVEF (Figure 7A), higher LVEDD (Figure 7B), as indicators for left ventricular dysfunction, and decreased TAPSE (Figure 7C) and an increase in systolic pulmonary artery pressure (Figure 7D) as indicators for right ventricular dysfunction.

## 3. Methods

### 3.1. Animal Model

All animal experiments were performed after approval (AZ 84-02.04.2012.A277), according to the European legislation and according to the FELASA guideline for the care and use of experimental animals. Eight-week-old apoE^+/+^ (Sprague Dawley, Charles River, Cologne, Germany) and apoE^−/−^ (SDapoEtm1sage rats; product number: TGRA3710HM9; Compor Zr ^®^ Zinc-finger nuclease target site: 5′ CAGGCCCTGAACCGAttctggGATTACCTGCGCTGGG; NCBI GeneID: NC_005100.2 GenBank; Sigma Aldrich Genetic Engineering Labs, Saint Louis, MO, USA) [8] rats were fed a normal diet (3.3% crude fat, cholesterol-free) or western type diet (WD, Ssniff Spezialdiaten, Soest, Germany) for 12 weeks. Rats were kept in standardized conditions (21 °C ± 2 °C, 60% ± 5% humidity, and a 12 h light/dark cycle, free access to food and water ad libitum).

Blood analysis, including the concentration of lipids (cholesterine, triglycerides), protein and glucose, uric acid, creatinine, as well as differential blood cell count, was performed at different time points after starting the diet.

At the end-point (after 20 weeks), rats were euthanized using cervical dislocation and were subjected to necropsy examination and the organs were immediately fixed in 4% neutral buffered formalin. The collected organs (heart, lung, liver, kidney, spleen, intestine) were excised, processed, embedded in paraffin, and cut into 5 µm thick sections. Sections were stained with Hematoxylin and Eosin (HE) and elastin van Gieson stain for the identification of elastic, muscles and collagen fibers and analyzed for different histological features using a light microscope.

### 3.2. Organ Scores

For the semiquantitative pattern characterization of organ structure alterations, we modified current scoring methods [9,10,11] to characterize specific features for each organ (Table 5): 1—none/negligible, 2—mild, 3—moderate, 4—moderate to severe, 5—severe. Whole sections were always evaluated using ×50, ×100 and ×400 magnification (5 different fields per organ) and analyzed by the same blinded observer. The average of all scores was considered.

### 3.3. Morphometric Analysis

For morphometric quantification, defined images of all hearts were obtained. Right and left ventricular wall thickness and ventricular cavity dimensions were quantified. Sirius red staining of the heart tissue was performed to quantify total collagen content in normal light and collagen type I and III in polarized light using Image J (National Institutes of Health, Bethesda, MD, USA).

Technical limitation: due to the fact that adequate equipment for measuring heart function in rats was missing, we limited our animal analysis to morphological parameters.

### 3.4. Study Population

A total of 120 Caucasian patients with heart failure (CM group) and 190 healthy outpatients (control group) were randomly selected between August 2017 and September 2019 from the Department of Cardiovascular Medicine of the University of Medicine and Pharmacy (Craiova, Romania) and included in the study. The local ethics committee approved the study protocol (No. 20/26.02.2016) and written informed consent was obtained before study enrollment from all patients.

The inclusion criteria included: (1) left ventricular ejection fraction <45% and/or left ventricular shortening fraction <25%; left ventricular end-diastolic diameter >3.2 cm/m^2^ (>117% of normal value) and/or left ventricular end-diastolic volume >75 mL/m^2^, measured by transthoracic echocardiography; (2) a history of myocardial infarction or evidence of clinically significant (≥70% narrowing of a major epicardial artery) coronary artery disease (the ischemic CM = ICM); and (3) dilation and impaired contraction of left ventricle or both ventricles of idiopathic, familial/genetic, viral and/or immune, toxic origin, or associated with recognized cardiovascular disease in which the degree of myocardial dysfunction is not explained by normal loading conditions or the extent of ischemic damage (non-ischemic CM = DCM).

The exclusion criteria included: (1) acute myocardial infarction, unstable angina or percutaneous coronary intervention (PCI) within 90 days before screening, or CABG surgery within 180 days before screening; (2) previously underwent cardiac surgery with a remodeling procedure, left ventricular assist device placement or cardiomyoplasty; (3) cardiac resynchronization therapy (CRT) within 6 months (180 days) before screening; and (4) active myocarditis, constrictive pericarditis, restrictive, significant valvular heart disease, hypertrophic or congenital cardiomyopathy.

The healthy volunteers were recruited based on the following criteria: (1) no symptoms of cardiovascular dysfunction, (2) normal ECG, and (3) no history of chronic disease. Clinical parameters (body mass index, ECG and smoking habit) for each participant were collected.

Collected data were (1) anthropometric data (age, gender, weight, height, body surface area (BSA), body mass index (BMI)); (2) clinical data (time since diagnosis, blood pressure, heart rate, heart failure treatment, complete physical examination); (3) rest 12 leads electrocardiogram (cardiac rhythm, heart rate and potential conduction/rhythm disturbances); (4) transthoracic echocardiography (conventional measurement for left and right ventricles dimensions and function); (5) laboratory analysis (total blood count, total cholesterol, LDL cholesterol, HDL cholesterol, triglycerides, uric acid, urea, creatinine, alanine aminotransferase, aspartate aminotransferase, glycemia, erythrocyte sedimentation rate, blood sample for DNA extraction).

### 3.5. Echocardiography

Routine echocardiographic examinations were performed at the echocardiography core facility [12,13]. The patients were examined in the left lateral decubitus position by using the dedicated ultrasound equipment (GE-Vivid S6, Avante Health Solution, Concord, NC, USA) with a 3.5 MHz transducer. Conventional measurements for left and right ventricles dimensions and function were obtained from standard views: the parasternal long axis view, short axis views and apical views recorded with a frame rate of a minimum of 55 frames per second. Valvular heart disease was assessed by combining Color Doppler echocardiography with spectral Doppler measurements. The LVEF was calculated in the apical four- and two-chamber views using the modified biplane Simpson’s rule. Further measurements included the left ventricular end-diastolic diameter (LVEDD), the left ventricular end-systolic diameter (LVESD), the septum thickness, the right-ventricular diastolic diameter (RVDD), the tricuspid regurgitation pressure gradient (TRPG), the tricuspid annular plane systolic excursion (TAPSE) and the pulmonary arterial pressure (PAPs). All echocardiographic measurements were performed following current recommendations for chamber quantification [14] at the specialized laboratory within the Department of Cardiology, University of Medicine and Pharmacy, Craiova, Romania. Image acquisition was carried out by a single expert examiner. Stored data were analyzed offline using EchoPac version BT13 (GE Vingmed Ultrasound, Horten, Norway) dedicated software and all measurements were performed by a single experienced reader blinded about the subject’s information.

### 3.6. Laboratory Analysis

Blood chemistry levels: total blood count, concentration of total cholesterol, LDL cholesterol, HDL cholesterol, triglycerides, uric acid, urea, creatinine, alanine aminotransferase, aspartate aminotransferase, glycemia and erythrocyte sedimentation rate, were performed at the hospital laboratory facility. In addition, blood samples from all participants were collected in tubes containing ethylenediaminetetraacetic acid (EDTA) and transported to the genetic human laboratory. After centrifugation, serum samples were stored at −80 °C until analysis while the sediment was stored at −20 °C until DNA extraction.

### 3.7. DNA Extraction

Genomic DNA was extracted from 300 μL of peripheral centrifuged blood. The isolation procedure was performed using a Wizard ^®^ Genomic DNA Purification Kit (Promega, Madison, WI, USA). Approximately 100 μL of DNA was obtained at the end of the procedure. The purity and concentration of DNA were measured by a Nanodrop 2000 (Spectrophotometer—Eppendorf Biophotometer, Wien, Austria) with DNA concentrations (mean = 112 ± 9.8 ng/μL) and optical density ratios (at 260/280 nm) (mean = 1.9 ± 0.2). Isolated DNA samples were preserved and stored at −20 °C until genotyping assessments were conducted.

### 3.8. Genotyping

The analysis of *apoE* genotype variants was performed using the protocol from Main BF et al. [15] that we adapted. Regarding oligonucleotides, allele-specific oligonucleotide primers were designed and synthesized according to the previous literature. Each primer was specific for the single base change that resulted in either Cys or Arg at positions 112 and 158 (Table 6).

PCR amplification was performed in an automated thermocycler (Thermocycler Eppendorf Mastercycler, Darmstadt, Germany) using the Promega kit GoTaq ^®^ DNA Polymerase for PCR. The total volume in each PCR reaction was 22 µL, including 5X GoTaq ^®^ Green, MgCl2 (50 mmol/L), dNTP (10 mmol/L), primers (20 pmol), template DNA, GoTaq ^®^ DNA Polymerase (4U) and ddH2O. Each PCR reaction contained the common primer (primer H) and one of the allele-specific primers. Therefore, four reaction mixtures were required per subject for genotype diagnosis. PCR reaction condition: after an initial DNA denaturation at 96 °C, 40 cycles of denaturation were performed at 96 °C, annealing at 58 °C and extending at 65 °C.

After PCR amplification, the samples were run on a horizontal electrophoresis system for 40 min, at 150w/150A over 2% agarose gel stained with Peqgreen to highlight the bands of the DNA. A CCD camera system (G:Box Chemi HR@Syngene, Cambridge, UK) was used to visualize the banding patterns. The banding patterns were then compared to the known apoE primer combination of every genotype (Figure 8 and Table 7).

### 3.9. Statistical Analysis

Data represent the mean ± SEM. Statistical analysis was performed with Prism 6.1 software (GraphPad). Means of two groups were compared with unpaired Student’s *t*-test, more than two groups with 1-way ANOVA followed by Tukey’s post hoc-test, and more than 2 variables with 2-way ANOVA and Tukey’s multiple comparison test, as indicated. *p*-values of <0.05 were considered significant.

Baseline characteristics were assessed by standard descriptive statistics. Continuous variables are presented as the mean ± standard deviation (SD). Categorical variables are presented as a frequency and percentage. Statistical differences between continuous variables were determined with Student’s *t*-test or with one-way ANOVA, followed by Tukey’s Multiple Comparisons testing, and statistical differences between categorical variables were determined with Chi Square test.

Individuals homozygous for the common genotypic variants were used as the reference to test for any association by calculating the odds ratio (OR) with a 95% confidence interval (CI). Statistical analysis was performed with SPSS 22.0 (SPSS, Armonk, NY, USA). A *p*-value of <0.05 was regarded as statistically significant.

## 4. Discussion

In this study, we showed that apoE-deficient rats present multiple organ damages besides the known predisposition for obesity and affected lipid metabolism. They showed increased retention parameters, mostly after the high-fat diet, and kidney injuries, such as vacuolization, congestion and glomerular thickening. The liver showed also signs of hepatocellular vacuolization, fat deposition and focal necrosis in apoE^−/−^ rats, the lung showed more edema and thickness of the alveolar wall, while the spleen showed more lymphoid hyperplasia in apoE^−/−^ compared with wild-type rats.

Heart tissue showed significant morphological changes in apoE^−/−^ rats, mostly after the high-fat diet, such as loose interstitial tissue, focal necrosis and edema. Interestingly, the right ventricle and not the left ventricle of apoE^−/−^ rats fed the high-fat diet showed increased bundles of thin tissue and mononuclear cell infiltration. The dimensions of both ventricles were not different; however, the collagen content seemed to only be significantly affected by apoE deficiency in the right ventricle. Similarly, the collagen1/collagen3 ratio was only increased in the right ventricle of apoE^−/−^ rats, suggesting significant changes in the extracellular matrix in the right ventricle compared with the left ventricle.

In patients, we were able to find a correlation between the presence of the ε4 allele and cardiomyopathy, but not with the etiology of the cardiomyopathy (ischemic or dilatative). While Jurkovicova et al. found ε2/ε4 as the predominant genotype in patients with dilated cardiomyopathy [6], we found ε3/ε4 to be the predominant genotype in our Romanian patients with ischemic or dilatative cardiomyopathy. The possible mechanisms for inducing severe forms of cardiomyopathy in the ε4 allele carriers are not known and need to be investigated. Interestingly, we found also a significant association between right ventricle dysfunction and the presence of the ε4 allele in the patients, which was not reported until now.

The missing or dysfunctional apoE protein, such as apoE4, is always associated with increased intracellular cholesterol and lipid accumulation and accompanied by cardiomyopathy and cardiac diastolic dysfunction [16,17,18]. Repetitive inflammatory and apoptotic processes are considered as the mechanisms that firstly modify the structure and then subsequently the function of the heart, being the motor of heart failure progression [19]. We can speculate that the proinflammatory [20] and proapoptotic [21] effects of apoE4 that had been reported could have a possible involvement in the progression of cardiomyopathy. For example, a possible mechanism may involve Cyclophilin A. The presence of apoE4 activates a pro-inflammatory signaling pathway in pericytes, including Cyclophilin A–nuclear factor-κB–matrix-metalloproteinase-9 (MMP9), leading to the destruction of the blood–brain barrier, with neuronal dysfunction [22]. Additionally, in the heart, Cyclophilin A is the main ligand for extracellular matrix metalloproteinase inducer (EMMPRIN), being associated with the development of cardiomyopathy and heart failure in both patients and animal models [23]. However, the exact interference between Cyclophilin A and apoE in cardiovascular diseases requires further investigation.

Furthermore, apoE was found to be significantly increased in apoptotic bodies [24], which implies a role of apoE in the clearance of apoptotic bodies and dead tissue, like after myocardial infarction. Moreover, it is known that apoE is a potent inhibitor of cell proliferation and has effects on modulating angiogenesis [25] and is also involved in the repair response to tissue injury with antioxidant activity [26]. Thus, the presence of a dysfunctional apoE, such as apoE4, may impair the recovery after an injury and an inflammatory process.

The architecture of the heart is made to support a huge biomechanical burden. The fibroblasts are responsible for sustaining the cardiomyocytes’ function but also for the qualitative and quantitative maintenance of the extracellular matrix. Under physiological conditions, fibroblasts are arranged in layers and strands, and are elongated and flattened cells that border on cardiomyocytes in an endomysial collagen network. This distribution model suggested from the very beginning the fundamental role of all the structures of the heart in taking over the tensile forces that appear at the parietal level [27]. Moreover, fibroblasts are actively involved in all repairing processes in the heart [28,29,30]. Thus, affecting fibroblasts’ integrity and function represents an important factor for developing heart failure [28,29]. Whether apoE can affect cardiac fibroblasts and subsequently contribute to the development of dilated cardiomyopathy must be further investigated. It is known that in human fibroblasts, apoE4 binds to the SirT1, MADD, or COMMD6, known to regulate transcription by the NFκB complex, triggering the inflammatory and apoptotic pathways [31].

Recently, more studies are emphasizing a major effect of lipoproteins on cardiac function independent of coronary artery disease. For example, cholesterol-lowering therapy can reduce oxidative stress and decrease levels of tumor necrosis factor alpha, thus counteracting structural and metabolic remodeling, enhancing cardiac function [32], attenuating pressure overload-induced heart failure [33] or even preventing heart failure with preserved ejection fraction [34].

Interestingly, it seems that there is a difference between the left and right ventricle fibroblasts. In the absence of injury, right and left ventricular fibroblasts appear to have similar characteristics. However, the right ventricle has unique characteristics that make it susceptible to certain genetic conditions associated with interstitial remodeling, such as arrhythmogenic right ventricle cardiomyopathy. Other studies showed that the perturbation of the collagen1/collagen3 ratio may be responsible for the increase in right ventricular stiffness [35,36,37]. The cellular composition of the right ventricle has not been systematically studied and the molecular links between genetic alterations and right ventricular interstitial remodeling remain unknown [35].

In the present study, we observed that the right ventricle of apoE^−/−^ rats presented more thin tissue bundles and mononuclear cell infiltration and an increased collagen1/collagen3 ratio, while these factors remained unchanged in the left ventricle. Similarly, the ε4 allele carriers showed significantly higher right ventricular dysfunction. Recently, some studies showed that apoE deficiency appears to be a novel risk factor for pulmonary artery hypertension (PAH) and patients with PAH have reduced expression of apoE in their lungs [38,39]. Additionally, apoE^−/−^ mice develop atherosclerosis with severe PAH when fed a high-fat diet (HFD) and have increased levels of endothelin (ET)-1. Atherosclerosis and PAH have the same pathophysiological processes, such as endothelial dysfunction and increased inflammatory mediators such as IL-6 [40]. This could be a mechanism of right ventricular dysfunction independent of the left ventricle failure.

## 5. Limitations

Besides technical limitations regarding the missing equipment for performing echocardiography in rats, we can consider the absence of more advanced imaging such as Speckle-tracking echocardiography or cardiac MRI that would enable the better analysis of the right and left ventricular dysfunction as a limitation. However, the acquired conventional echocardiography data were enough to provide solid data and significant correlations for the current study. The small population size most likely did not affect the results because the presence of the ε4 allele was significantly higher in the CM population, similar to other cohorts. Further, our study was more of an observational study based on a case–control design. However, since this is the first article reporting the relation between the presence of the ε4 allele and right ventricular dysfunction, the results should be further validated in a specific risk population.

## 6. Conclusions

Our results confirm that the presence of the ε4 allele is an independent risk factor for heart failure, independent of etiology and lipid metabolism. The underlying mechanisms remain to be elucidated. Moreover, we show for the first time that the presence of the ε4 allele is associated with right ventricle dysfunction, implying different regulatory mechanisms of fibroblasts and extracellular matrix in both ventricles. It is essential that this is considered and thoroughly investigated before the design of therapeutical strategies for patients with heart failure.

## Figures and Tables

**Figure 1 ijms-22-09688-f001:**
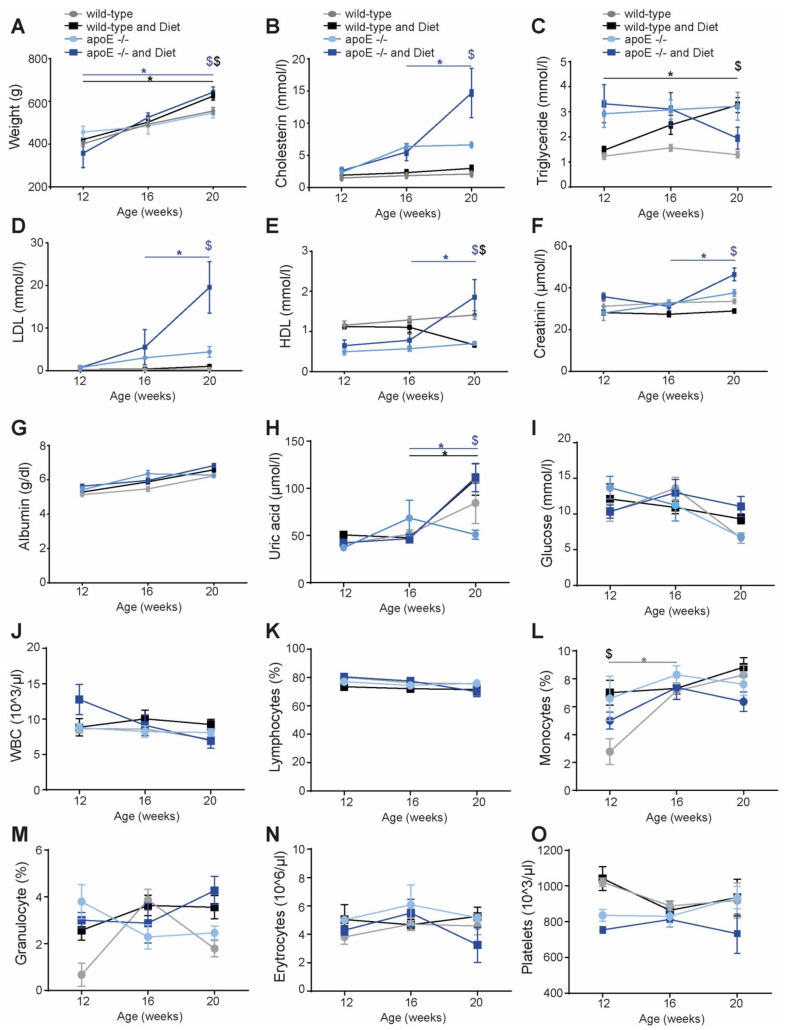
Blood analysis of apoE^−/−^ rats at different time points under feeding high-fat diet (**A**–**O**). Indicated parameters and cell components were determined after 4 weeks (age 12 weeks), 8 weeks (age 16 weeks) and 12 weeks (age 20 weeks), respectively (*n* = 5–15), under normal diet (bright blue for apoE^−/−^ and grey for wild-type rats) or high-fat diet (dark blue for apoE^−/−^ and black for wild-type rats). * *p* < 0.05 vs. earlier time points, ^$^
*p* < 0.05 high-fat vs. normal diet (in black for wild-type and in blue for apoE^−/−^ rats).

**Figure 2 ijms-22-09688-f002:**
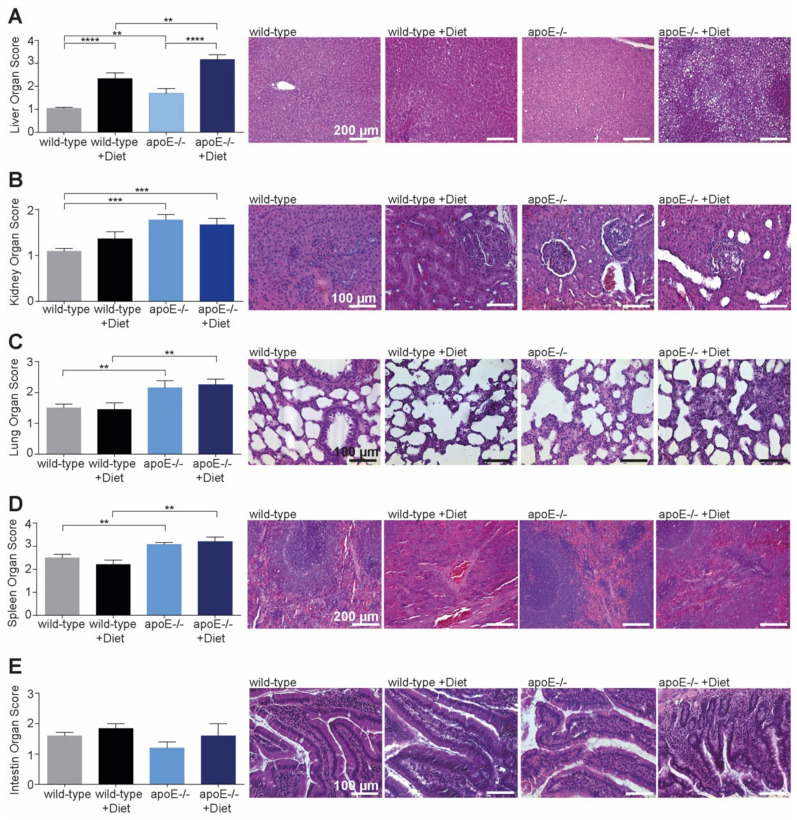
Changes in organ morphology of apoE^−/−^ rats after feeding high-fat diet (**A**–**E**). Twelve weeks (age 20 weeks) after feeding normal or high-fat diet, apoE^−/−^ rats showed pathological changes in almost all organs compared with wild-type rats. While pathological changes in kidney, lung, and spleen were not related to diet, the liver was dramatically affected by high-fat diet, especially in apoE^−/−^. ** *p* < 0.0, *** *p* < 0.001, **** *p* < 0.0001, *n* = 5–15, scale bar is indicated in the first image of each group.

**Figure 3 ijms-22-09688-f003:**
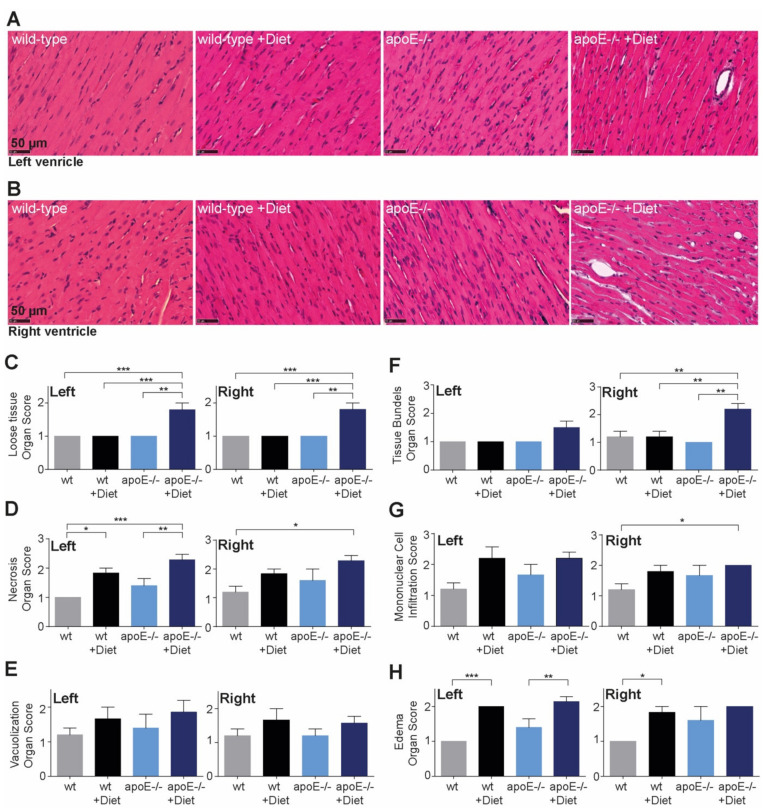
Changes in heart morphology of apoE^−/−^ rats after feeding high-fat diet Twelve weeks (age 20 weeks) after feeding normal or high-fat diet, apoE^−/−^ rats showed differentiated pathological changes in left and right ventricle compared with wild-type rats (**A**–**H**). While interstitial tissue of each ventricle was equally affected and presented necrosis foci, the right ventricle showed more thin tissue bundles and mononuclear cell infiltration and the left ventricle, more edema in apoE^−/−^ rats fed with high-fat diet. **p* < 0.5, ** *p* < 0.01, *** *p* < 0.001, *n* = 5–15, scale bar is indicated in the first image of each group.

**Figure 4 ijms-22-09688-f004:**
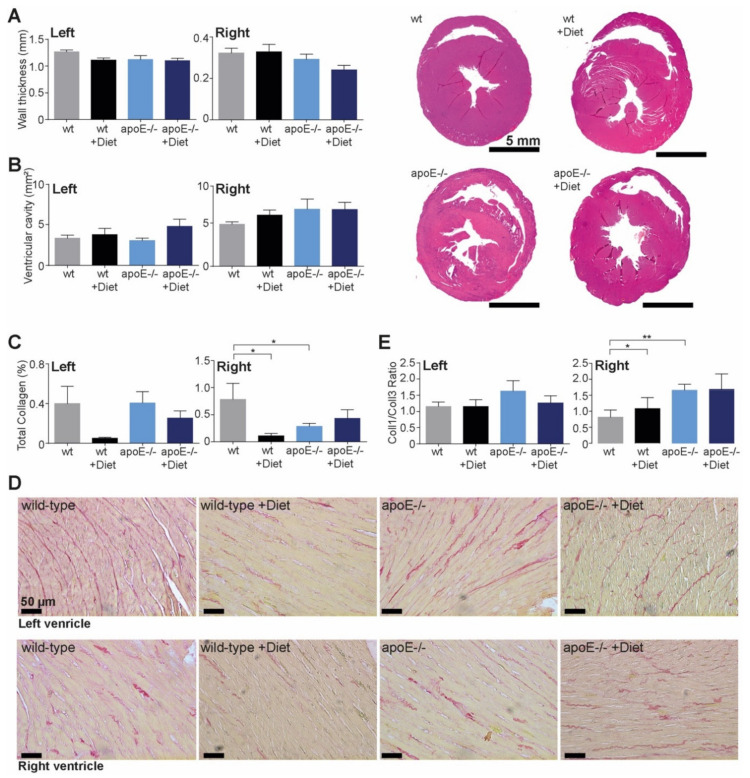
Histomorphometrical characterization of hearts in apoE^−/−^ rats fed normal or high-fat diet (**A**–**D**). Twelve weeks (age 20 weeks) after feeding normal or high-fat diet, apoE^−/−^ rats showed no differences in the heart dimensions compared with wild-type rats, while the right ventricle showed a decrease in collagen deposition and increased collagen1/collagen3 ratio in apoE^−/−^ rats or rats fed high-fat diet. * *p* < 0.5, ** *p* < 0.01, *n* = 5–15, scale bar is indicated in the first image of each group.

**Figure 5 ijms-22-09688-f005:**
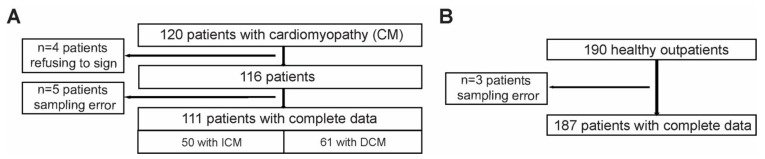
Study design. Patients with cardiomyopathy (**A**) and the healthy volunteers (**B**) recruited for the current study.

**Figure 6 ijms-22-09688-f006:**
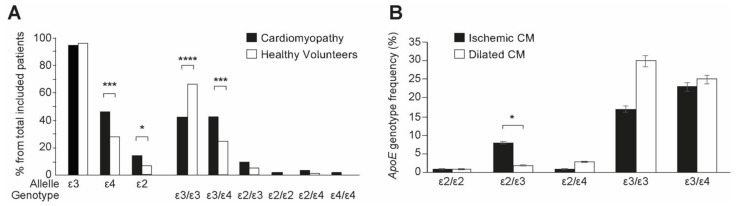
Genotype frequency and allele frequency between the groups. (**A**) Patients with cardiomyopathy (black columns) present ε2 and ε4 alleles more frequently as compared with controls (white columns). (**B**) The distribution of different alleles was not dependent on ischemic (black columns) or non-ischemic (white columns) etiology of cardiomyopathy (* *p* < 0.5, *** *p* < 0.001; **** *p* < 0.0001).

**Figure 7 ijms-22-09688-f007:**
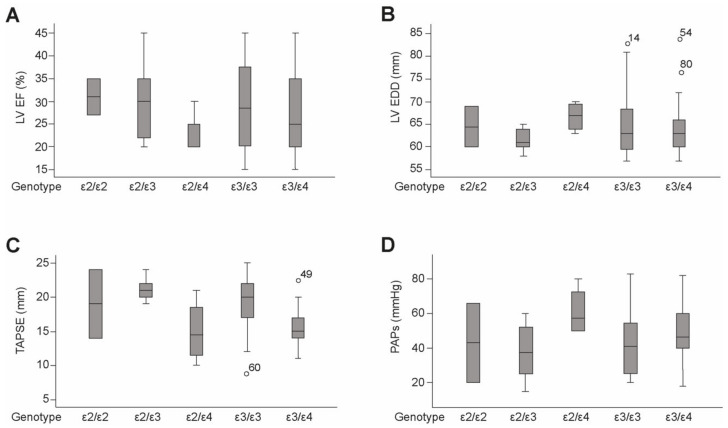
Functional parameters in different genotypes. (**A**) Left ventricular ejection fraction (LVEF), (**B**) left ventricular end-diastolic diameter (LVEDD), (**C**) tricuspid annular plane systolic excursion (TAPSE) and (**D**) and pulmonary arterial pressure (PAPs).

**Figure 8 ijms-22-09688-f008:**
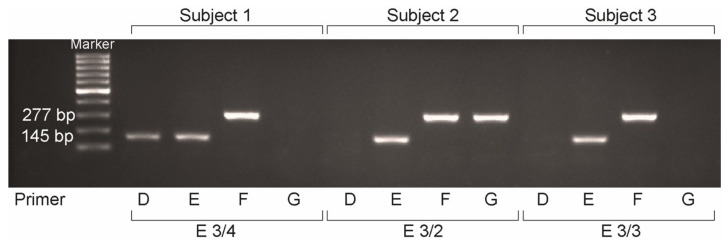
Apolipoprotein E genotyping using the polymerase chain reaction and allele specific oligonucleotide primers. Peqgreen-stained agarose gel showing the products from the PCR reactions. Lanes 1 through 4 are the reactions for subject 1: lane 1 is positive (primer D), lane 2 is positive (primer E), lane 3 is positive (primer F), and lane 4 is negative (primer G) indicating the E 3/4 genotype. Lanes 5 through 8 are the reactions for subject 2. Lane 5 (primer D) is negative, indicating the absence of the e4 allele; lanes 6 through 8 are positive with primers E, F, and G, indicating the presence of E3 and E2 yielding a genotype of apoE 3/2. Lanes 9 through 13 are the reactions for subject 3: lane l is negative (primer D), lane 2 is positive (primer E), lane 3 is positive (primer F), and lane 4 is negative (primer G) indicating the E 3/3 genotype.

**Table 1 ijms-22-09688-t001:** Demographics and clinical characteristics of control and patient groups.

Variable	CM Patients *n* = 111	Controls *n* = 187	*p* Value
Age (years)	66 ± 8.68	57 ± 10.3	*p* < 0.01
Sex (male)	86 (77.48%)	131 (70.06%)	N
BMI, kg/m^2^	27.78 ± 4.90	26.56 ± 3.89	N
Alcohol, *n* (%)	13 (11.71%)	33 (17.65%)	N
Smoking, *n* (%)	29 (26.13%)	95(50.8%)	*p* < 0.01

**Table 2 ijms-22-09688-t002:** The *apoE* genotype in CM patients and control group.

	CM Patients (*n* = 111)	Control (*n* = 186)	*p*-Value	Odds Ratio	95% Confidence Interval
Allelle	Allellic count (%)	Allellic count (%)			
ε3	105 (94.59%)	180 (96.25%)	0.35	0.58	0.183–1.855
ε4	52 (46.85%)	53 (28.49%)	0.001	2.21	1.355–3.611
ε2	16 (14.41%)	13 (6.9%)	0.03	2.24	1.034–4.857
Genotype	*n* (%)	*n* (%)			
ε3/ε3	47 (42.34%)	123 (66.12%)	0.0001	0.38	0.231–0.610
ε3/ε4	48 (43.24%)	46 (24.73%)	0.0009	2.31	1.404–3.831
ε2/ε3	10 (9%)	9 (4.8%)	0.15	0.51	0.202–1.306
ε2/ε2	2 (1.8%)	1 (0.5%)	0.29	0.29	0.026–3.287
ε2/ε4	4 (3.6%)	3 (1.6%)	0.27	2.28	0.501–10.383
ε4/ε4		4 (2.1%)	0.11		

**Table 3 ijms-22-09688-t003:** The apoE genotype based on CM etiology.

	ICM (*n* = 50)	DCM (*n* = 61)	*X* ^2^	*p*-Value
Allelle	Allellic count (%)	Allellic count (%)		
ε3	48 (96%)	57 (93.44%)	0.315	0.554
ε4	24 (48%)	28 (45.9%)	0.485	0.825
ε2	10 (20%)	6 (9.8%)	2.537	0.111
Genotype	*n* (%)	*n* (%)		
ε3/ε3	17 (34%)	30 (49.18%)	2.593	0.108
ε3/ε4	23 (46%)	25 (40.98%)	0.499	0.481
ε2/ε3	8 (16%)	2 (3.28%)	5.425	0.019
ε2/ε2	1 (2%)	1 (1.64%)	0.021	0.886
ε2/ε4	1 (2%)	3 (4.91%)	0.674	0.412

**Table 4 ijms-22-09688-t004:** The main characteristics of the patient according to ε4 presence.

	ε4 Present (*n* = 52)	ε4 Absent (*n* = 59)	*p*-Value
Age, years (Range)	67.8 (40–93)	64.35 (28–88)	0.1250
Male gender, *n* (%)	42 (80.8%)	44 (74.6%)	0.0910
BMI, kg/m^2^	26.56 ± 3.89	28.46 ± 5.60	0.5684
Risk Factors			
Alcohol, *n* (%)	3 (5.7%)	10 (16.9%)	N
Smoking, *n* (%)	11 (21.1%)	18 (30.5%)	N
Diabetes, *n* (%)	23 (44.2%)	20 (33.9%)	N
NYHA classification			
1	3 (5.8%)	6 (10.2%)	N
2	2 (3.8%)	7 (11.9%)	N
3	30 (57.7%)	30 (50.8%)	N
4	17 (32.7%)	16 (27.1%)	N
DCM etiology			
Non-ischemic	23 (44.2%)	28 (47.5%)	0.7335
Ischemic	29 (55.8%)	31 (52.5%)	0.7335
ECG			
Sinus rhythm	24 (46.2%)	17 (28.8%)	0.0589
Atrial fibrillation	23 (44.2%)	32 (59.3%)	0.2927
Other	5 (9.6%)	10 (16.9%)	0.2594
LBBB	11 (21.2%)	20 (33.9%)	0.1353
LVEF			
35–45%, *n* (%)	4 (7.7%)	14 (23.7%)	0.0221
≤35%, *n* (%)	48 (92.3%)	42 (76.3%)	0.0045
LVEF mean ± SD (%)	26.67 ± 8.45	29.57 ± 9.92	0.0992
Echocardiography			
LVEDD (mm)	64 ± 5.27	64.23 ± 6.11	N
LVESD (mm)	52.27 ± 8.19	52.27 ± 7.7	N
Septum (mm)	11.14 ± 2.18	11.05 ± 1.94	N
RVDD (mm)	41.52 ± 7.82	42.27 ± 6.76	N
TRPG (mmHg)	37.04 ± 13.94	29.95 ± 14.31	0.0096
TAPSE (mm)	15.3 ± 2.63	19.8 ± 3.58	<0.0001
PAPs (mmHg)	50.44 ± 16.47	40.68 ± 15.94	0.0019
Blood parameters			
Total cholesterol, mg/dL	137.73 ± 35.32	134.24 ± 36.53	0.6110
Creatinine, mg/dL	1.29 ± 0.85	1.19 ± 0.56	0.4609
Estimated GFR, mL/min/1.73 m^2^	73.05 ± 32.61	73.98 ± 30.68	0.88400
Sodium, mmol/L	137.59 ± 3.84	136.58 ± 5.64	N
Potassium, mmol/L	4.66 ± 0.67	4.59 ±0.8	N

**Table 5 ijms-22-09688-t005:** Histological tissue features and organ scores.

**Heart**	Loose tissue composition Vacuolization of myocardium Thin tissue bundles Edema Focal inflammation (Mononuclear cells infiltration) Focal necrosis	Scores: 1—none/negligible 2—mild 3—moderate 4—moderate to severe 5—severe
**Liver**	Hepatocellular vacuolization Congestion Focal inflammation Focal necrosis	
**Kidney**	Vacuolization of glomerular epithelium microcysts Congestion Increased cellularity of glomeruli Glomerular thickening	
**Lung**	Edema Congestion Thin blood vessels Thickened alveolar wall	
**Spleen**	Lymphoid hyperplasia Congestion	
**Intestine**	Edema Increased cellularity of lamina propria Fusion of villi	

**Table 6 ijms-22-09688-t006:** Oligonucleotide primer sequences: primer sequence (*5′–3′*). Primers D and E are similar; their differences are shown below; the same goes for primer F and G. Primer H is the common primer.

Primer D	TACTGCACCAGGCGGCCTCG
Primer E	TACTGCACCAGGCGGCCTCA
Primer F	GCCTGGTACACTGCCAGTCG
Primer G	GCCTGGTACACTGCCAGTCA
Primer H	AAGGAGTTGAAGGCCTACAAAT

**Table 7 ijms-22-09688-t007:** The primer combination of every genotype. The E3/3 genotype yielding a 145 base pair product with primer E, indicating the presence of Cys 112, and a 277 base pair product with primer F indicating the presence of Arg 158. The heterozygote E3/4 produces appropriately sized products with primers E and F identifying the E3 allele, but also produces a 145 base pair product with primer D indicating Arg 112. Homozygous E4/4 subjects only react with primers D and F, marking the presence of Arg 112 and Arg 158, while E2/2 homozygotes only react with primers E and G, indicating Cys 112 and Cys 158.

Genotype	Primer D	Primer E	Primer F	Primer G
E 2/2	−	+	−	+
E 2/3	−	+	+	+
E 3/3	−	+	+	−
E 3/4	+	+	+	−
E 4/4	+	−	+	−
E 2/4	+	+	+	+

## Data Availability

The data that support the findings of this study are available from the corresponding author on reasonable request.

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
