# Peer review of "Apolipoprotein E4 Is Associated with Right Ventricular Dysfunction in Dilated Cardiomyopathy—An Animal and In-Human Comparative Study"

_ijms, 2021, doi:10.3390/ijms22189688_

Round 1

Reviewer 1 Report

To:

Editorial Board

Title: “Apolipoprotein E4 is associated with right ventricular dysfunc-tion in dilated cardiomyopathy - An animal and in-human com-parative study”

Dear Editor,

I read this paper and I think that:

  • Please include more numerical data in the results section of the abstract. Please provide.
  • Inter and intraobserver variability coefficients for echocardiographic evaluations should be performed.

Author Response

I read this paper and I think that:

  1. Please include more numerical data in the results section of the abstract. Please provide.

We have now added numerical data in the abstract, in the result section. However, since there are a lot of groups to compare and a limited number of words allowed in the abstract, we were able to present only relevant data (see Abstract, page 1): “Methods and Results: We have shown that apoE deficient rats present multiple organ damages (kidney, liver, lung and spleen) besides the known predisposition for obesity and affected lipid metabolism (2-fold increase in tissular damages in liver and 1-fold increase in kidney, lung and spleen). Heart tissue also showed significant morphological changes in apoE-/- rats, mostly after a high-fat diet. Interestingly, the right ventricle of apoE-/- rats fed a high-fat diet showed more damage and affected collagen content (~60% less total collagen content and double increase in collagen1/collagen 3 ratio) compared with the left ventricle (no significant differences in total collagen content or collagen1/collagen3 ratio). In patients, we were able to find a correlation between the presence of ε4 allele and cardiomyopathy (χ2=10.244; p=0.001), but also with right ventricle dysfunction with decreased TAPSE (15.3±2.63 mm in ε4-allele presenting patients vs. 19.8 ± 3.58 mm if ε4-allele absent, p<0.0001*) and increased in systolic pulmonary artery pressure (50.44±16.47 mmHg in ε4-allele presenting patients vs. 40.68±15.94 mmHg if ε4-allele absent, p=0.0019).”

  1. Inter and intraobserver variability coefficients for echocardiographic evaluations should be performed.

We are grateful to the referee for this observation. We have now added a supplementary statement in our method part (see Methods, page 4, paragraph 4): “All echocardiographic measurements were performed following current recommendations for chamber quantification (reference no.14) at the specialized laboratory within the Department of Cardiology, University of Medicine and Pharmacy, Craiova, Romania. Image acquisition was done by a single expert examiner. Stored data were analyzed offline using EchoPac version BT13 (GE Vingmed Ultrasound, Horten, Norway) dedicated software and all measurements were performed by a single experienced reader blinded about the subject’s information.”

Reviewer 2 Report

This is a very interesting study combining animal and patient data. The style of the paper is very good. There is logical flow of data.

  1. The association in patients is a preliminary finding due to the case-control design. This limitation should be emphasized.
  2. It is not clear whether the effect of apo E is lipid-independent. There are major effects of lipoproteins on cardiac function independent of coronary artery disease. This point should also be emphasized. (see e.g.Aging (Albany NY). 2019 Sep 4;11(17):6872-6891. doi: 10.18632/aging.102218. Epub 2019 Sep 4.PMID: 31484164 Int J Mol Sci. 2019 May 6;20(9):2222. doi: 10.3390/ijms20092222.PMID: 31064116 Mol Ther. 2017 Nov 1;25(11):2513-2525. doi: 10.1016/j.ymthe.2017.07.017. Epub 2017 Aug 1.PMID: 28822689

Author Response

This is a very interesting study combining animal and patient data. The style of the paper is very good. There is logical flow of data.

We thank very much this referee for appreciation of our work, and we hope now that we have responded to all remaining concerns.

  1. The association in patients is a preliminary finding due to the case-control design. This limitation should be emphasized.

We thank for pointing this out. We have now added this as a limitation in our limitation section (see page 16, second paragraph): “Further, our study was more an observational study based on case-control design. However, since this is the first article reporting the relation between the presence of ε4 allele and right ventricular dysfunction, the results should be further validated in a specific risk population.”

  1. It is not clear whether the effect of apo E is lipid-independent. There are major effects of lipoproteins on cardiac function independent of coronary artery disease. This point should also be emphasized.

We are grateful for this comment. We have now emphasized and discussed these lipoprotein’s effects on our revised manuscript (see Discussion, page 15, paragraph 4): “Recently more studies are emphasizing a major effect of lipoproteins on cardiac function independent of coronary artery disease. For example, cholesterol-lowering therapy can reduce oxidative stress and decreased levels of tumor necrosis factor alpha, thus counteracting structural and metabolic remodeling, enhancing cardiac function (reference no.32), attenuating pressure overload-induced heart failure (reference no.33), or even preventing heart failure with preserved ejection fraction (reference no.34)”